# Do B Vitamins Enhance the Effect of *Omega*-3 Polyunsaturated Fatty Acids on Cardiovascular Diseases? A Systematic Review of Clinical Trials

**DOI:** 10.3390/nu14081608

**Published:** 2022-04-12

**Authors:** Jie Zhu, Peng-Cheng Xun, Marissa Kolencik, Ke-Feng Yang, Alyce D. Fly, Ka Kahe

**Affiliations:** 1Nutrition and Foods Program, School of Family and Consumer Sciences, Texas State University, 601 University Drive, San Marcos, TX 78666, USA; j_z151@txstate.edu (J.Z.); mgk20@txstate.edu (M.K.); 2Department of Epidemiology and Biostatistics, School of Public Health-Bloomington, Indiana University, 1025 E. 7th Street, Bloomington, IN 47405, USA; pxun@indiana.edu; 3Department of Nutrition, Shanghai Jiao Tong University School of Medicine, 227 South Chongqing Road, Shanghai 200025, China; ykf789@163.com; 4Department of Applied Health Science, Indiana University School of Public Health-Bloomington, 1025 E. 7th Street, Bloomington, IN 47405, USA; afly@indiana.edu; 5Department of Obstetrics and Gynecology, Vagelos College of Physician and Surgeons, Columbia University Irving Medical Center, 622 West 168th Street, New York, NY 10032, USA; 6Department of Epidemiology, Mailman School of Public Health, Columbia University, 630 West 168th Street, New York, NY 10032, USA

**Keywords:** B vitamins, cardiovascular disease, *omega*-3 PUFAs, supplementation

## Abstract

Studies have suggested that B vitamins or *omega*-3 polyunsaturated fatty acids (PUFAs) may deter the development of cardiovascular disease (CVD). This systematic review aims to examine whether the combined supplementation of both B vitamins and *omega*-3 PUFAs could provide additional beneficial effects to prevent CVD beyond the effect of each supplement based on clinical trials published up to December 2021. The overall findings are inconsistent and inconclusive, yet the combined supplementation of these two nutrients may be more effective at reducing plasma homocysteine, triglyceride, and low-density lipoprotein-cholesterol than the individual components. The underlying mechanisms mainly include alleviating endothelial dysfunction, inhibiting atherosclerosis and lesion initiation, reducing oxidative stress, suppressing activation of pro-inflammatory cytokines, regulating endothelial nitric oxide synthase, and interfering with methylation of genes that promote atherogenesis. Although biologically plausible, the existing literature is insufficient to draw any firm conclusion regarding whether B vitamins can further enhance the potential beneficial effects of *omega*-3 PUFA intake on either primary or secondary prevention of CVD. The inconsistent findings may be largely explained by the methodological challenges. Therefore, well-designed high-quality trials that will use the combined supplementation of B vitamins and *omega*-3 PUFAs or dietary patterns rich in these two types of nutrients are warranted.

## 1. Introduction

Dietary intake of B vitamins or *omega*-3 polyunsaturated fatty acids (PUFAs) has been found to be inversely related to cardiovascular disease (CVD) [1,2]. The potential mechanisms behind this association are attributed to their roles in lowering the risk factors of CVD, including plasma homocysteine (Hcy), circulating triglycerides (TG) and C-reaction protein (CRP) [1,3,4,5]. These risk factors have been implicated in the atherosclerotic process [6] and endothelial dysfunction [7] which involve in CVD development, therefore, B vitamins and *omega*-3 PUFAs are expected to contribute to CVD prevention [8,9].

Nevertheless, a recent meta-analysis including 15 randomized controlled trials (RCTs) has found that B vitamins supplementations (vitamin B_6_, folate, or vitamin B_12_ given alone or in combination) have little impact on CVD prevention [10], although the other meta-analysis including five RCTs has showed a benefit for folic acid in CVD reduction [11]. Additionally, a meta-analysis including 79 RCTs also reported little to no effect of increasing *omega*-3 PUFAs intake (particularly in the form of supplements) on CVD prevention [12].

However, results from several clinical trials revealed a potential synergetic influence of B vitamins and *omega*-3 PUFAs to prevent CVD. In volunteers with normal or slightly to moderately increased blood lipids, supplementation of both B vitamins (vitamin B_6_ and folic acid) and fish oil for 4 weeks reduced the atherogenic index by 24%, which was greater than the decrease (12%) from fish oil supplementation alone [12]. Similarly, supplementation of both vitamin B_12_ and fish oil for 8 weeks lowered plasma Hcy concentration more than individual supplementation with either vitamin B_12_ or fish oil in healthy participants [3]. Since *omega*-3 PUFA and B vitamins are both involved in metabolism of one carbon units [6,13,14], alterations in each of these nutrients may influence Hcy concentration, methylation reactions and oxidative stress, all of which are involved in the pathological development of CVD [13]. Thus, co-ingesting these two nutrient groups may provide more beneficial effects to prevent CVD, compared to supplements of either B vitamins or *omega*-3 PUFAs alone.

Therefore, this review aims to examine whether the combined supplementation of B vitamins and *omega*-3 PUFAs could provide additional beneficial effects on improving risk factors to prevent CVD beyond the effects of either of them alone and delineate any inconsistencies and limitations of the current intervention studies, through comprehensive analysis of the available literature, and, lastly, provide suggestions for future research and dietetic practice or advice.

## 2. Methods

### 2.1. Search Strategy

We conducted the literature search on PubMed, EmBase, and the Cochrane Register of Controlled Trials up to December 2021 following the statement checklist of the PRISMA (Preferred Reporting Items for Systematic Reviews and Meta-Analyses) [15]. The current review was registered with PROSPERO, the international prospective register of systematic reviews (https://www.crd.york.ac.uk/prospero/, accessed on 1 December 2021), with the registration number CRD42018085993. Search terms included the word combinations of the following Medical Subject Headings: [“cardiovascular disease” or “cardiovascular disease risk factor” or “low-density lipoprotein cholesterol” or “LDL” or “C-reactive protein” or “CRP” or “triglyceride” or “TG” or “high-density lipoprotein cholesterol” or “HDL”] AND [“B vitamins” or “vitamin B_2_” or “riboflavin” or “vitamin B_6_” or “pyridoxine” or “folate” or “folic acid” or “vitamin B_12_” or “cobalamin” or “homocysteine” or “Hcy”] AND [“fish oil” or “*n*-3 fatty acids” or “*omega*-3 fatty acids” or “docosahexaenoic acid” or “DHA” or “eicosapentaenoic acid” or “EPA’’ or “alpha-linolenic acid” or “ALA”]. Reference lists of all identified articles and recently published reviews were also examined for the search terms.

### 2.2. Inclusion and Exclusion Criteria

Eligible studies were included in this review if they reported the synergistic effect of the combined supplementation of B vitamins (vitamin B2, vitamin B6, folic acid and vitamin B12) and fish oil (or an *omega*-3 PUFA term) on risk of CVD and/or its risk factors in humans. No selection restrictions were placed on investigations with respect to study type, study design, publication year or language (if an English abstract was provided). A large number of relevant publications were identified; however, the majority were excluded because they only reported the influence of either B vitamins or fish oil (or *omega*-3 PUFA) supplementation alone on the prevention or treatment of CVD. Thus, only studies investigating a synergistic effect of fish oil (or fatty acids) and B vitamins in humans were included. Finally, eligible articles were identified and assessed by one author (J.Z.). A second researcher (P.C.X.) independently evaluated and verified inclusion/exclusion decisions. Study selection and screening process shown in Figure 1.

### 2.3. Data Extraction and Study Quality Evaluation

Data were extracted from the original studies, which included first author name, the publication year, study location/design/duration, participants information, intervention, and outcomes. The quality criteria checklist of the Evidence Analysis Manual (2016) of the Academy of Nutrition and Dietetics was applied to evaluate study quality and risk of bias [16].

## 3. Results

The search yielded a total of 688 potentially relevant articles, of which 640 were excluded because they are review articles or animal studies or non-human intervention studies or with no abstract available; 48 full-text publications were retrieved and obtained for detailed review, of which 37 were excluded due to lack of related exposure or outcomes, and 4 additional articles were identified and included after reviewing the reference lists. Following detailed review, 15 studies met the inclusion criteria [3,4,5,6,12,14,17,18,19,20,21,22,23,24,25], i.e., reporting the synergistic effect of the combined supplementation of B vitamins (vitamin B_2_, vitamin B_6_, folic acid, and vitamin B_12_) and fish oil (or *omega*-3 fatty acids), or together with other nutrients in humans (Figure 1).

As shown in Table 1, the sample sizes ranged from 12 to 2501 participants with study duration ranging from 4 weeks to 4.7 years. Additionally, the dose ranges of the B vitamins or *omega*-3 PUFAs differed dramatically [vitamin B_6_ (2.5–80) mg/day, vitamin B_12_ (20–1000) μg/day, folic acid (150–10,000) μg/day and *omega*-3 PUFAs (0.2–2) g/day]. Fourteen studies [3,4,5,12,14,17,18,19,20,21,22,23,24,25] investigated the effects of the combined supplementation of B vitamins and *omega*-3 PUFAs on blood Hcy with most studies reporting Hcy-lowering effect [3,4,5,12,14,17,19,20,21,22,23,24,25]. The effects of the above combined nutrients on CRP were examined by eight trials, three of which reported beneficial effects [3,19,22] with insignificant changes in CRP values found in the other studies [4,5,18,20,21]. Among eleven trials investigating the impact of the combined supplementation of B vitamins and *omega*-3 PUFAs on plasma TG [3,4,5,12,17,18,19,20,21,22,24,25], seven studies reported the lowering effects of TG upon joint consumption of B vitamins and *omega*-3 PUFAs [3,4,12,18,19,21,25], while the four remaining trials demonstrated no change in plasma TG [5,20,22,24]. Additionally, seven trials investigated the impact of the combined administration of B vitamins and *omega*-3 PUFAs on plasma low-density lipoprotein cholesterol (LDL-C) with six studies [5,18,21,22,24,25] showing decreasing LDL-C effects and the other one reporting no obvious alteration of LDL-C concentration [23]. Moreover, five studies assessed the influence of the joint supplementation of B vitamins and *omega*-3 PUFAs on plasma high-density lipoprotein cholesterol (HDL-C) with elevating effects observed in two trials [12,18] and no changes reported in other three studies [5,22,24]. There is no intervention study included in this review reporting the effect of the combined B vitamins and *omega*-3 PUFAs administration on primary end points of CVD, because the trails were either conducted in participants with cardiovascular disease at baseline [6,14,17,22] or cardiovascular risk factors were only assayed in the studies [3,4,5,12,18,19,20,21,23,24,25].

For all included literature, 4 studies [19,21,24,25] received positive scores and 11 studies [3,4,5,6,12,14,17,18,20,22,23] received neutral scores by using the published format [26] based on the quality criteria checklist (Appendix A).

## 4. Discussion

Although several studies have demonstrated that the combined supplementation of B vitamins and *omega*-3 PUFAs has diverse effects on the CVD risk factors, it is insufficient to draw any firm conclusion as to whether B vitamins can further enhance the potential beneficial effects of *omega*-3 PUFA intake on either primary or secondary prevention of CVD.

### 4.1. B Vitamins and Omega-3 PUFAs Supplementation and CVD Risk Factors

Hyperhomocysteinemia is recognized as an independent risk factor of CVD [3,6,27]. Hcy can be either metabolized to cysteine or recycled to methionine with the aid of a group of B vitamins, through roles as essential coenzymes (vitamin B_2_, B_6_ and B_12_) or as an essential substrate donor (folate) in one-carbon unit metabolism [8]. It has been well established that supplementation with B vitamins efficaciously lowers plasma Hcy due to direct participation in degradation of Hcy [28,29].

*Omega*-3 PUFAs can originate from the plant sources, i.e., alpha-linolenic acid (ALA), and the more elongated and desaturated fatty acids from animal sources, i.e., eicosapentaenoic acid (EPA) and docosahexaenoic acid (DHA) [30]. *Omega*-3 PUFAs are the major fatty acids found in cholesteryl esters and phospholipids, which constitutes to the essential membrane component of cells and intracellular organelles [8,30]. However, the impact of *omega*-3 PUFAs supplementation on plasma Hcy remains elusive. Some studies have shown that *omega*-3 PUFAs consumption reduced plasma Hcy levels [3,31], whereas a significant decrease in plasma Hcy was not observed in other similar studies [18,32] These conflicting results may be due to interventions that differed in duration and used different types of participant populations [3].

Moreover, it was found that the combined supplementation of vitamin B_12_ (1000 μg) and fish oil (2 g, containing 980 mg DHA and 196 EPA) for 8 weeks was more effective at lowering plasma Hcy concentration than either vitamin B_12_ or fish oil supplementation alone in healthy participants [3]. Likewise, consumption of 500 mL/day milk (containing *omega*-3 PUFAs [EPA 1.4% and DHA 2.1% in milk fat], oleic acid [54.4% in milk fat], vitamins E [1.50 mg/100 mL], B_6_ [0.30 mg/100 mL], and folic acid [30 mg/100 mL]) decreased plasma Hcy levels in healthy participants [24], volunteers with mild hyperlipidemia [25], patients with peripheral vascular disease (PVD) [20], patients with metabolic syndrome (MS) [21], and participants suffering from myocardial infarction (MI) [22]. Additionally, a baked functional food product supplemented with sterols esters (1300 mg), fish oil (1000 mg EPA plus DHA), vitamins B_12_ (50 μg), B_6_ (2.5 mg), folic acid (800 μg), and coenzyme Q10 (3 mg) that was administered in a balanced diet decreased fasting Hcy in patients with mild-mixed hyperlipidemia [5]. A recent meta-analysis also suggested that a combination of *omega*-3 PUFAs (0.2–6.0 g/day), folic acid (150–2500 μg/day), and vitamins B_6_ and B_12_ appeared to be superior at reducing plasma Hcy than *omega*-3 PUFAs supplementation alone [27].

CRP is another independent risk predictor for CVD as a biomarker of systemic inflammation [22,33]. In this review, eight trials investigated the effects of the combined intake of B vitamins and *omega*-3 PUFAs on CRP (mainly in the form of high sensitivity CRP) [3,4,5,18,19,20,21,22]. Three studies reported beneficial effects on CRP [3,19,22], while the other studies failed to find significant changes in CRP values following the joint supplementation [4,5,18,20,21]. Huang et al. showed that administration of fish oil and vitamin B_12_ significantly reduced plasma CRP levels, compared to the values of the participants at baseline [3]. Consistent with this trial, Carrero et al. demonstrated that high plasma CRP concentrations declined in MI patients who consumed fortified semi-skimmed milk supplemented with *omega*-3 PUFAs, oleic acid, B vitamins (folic acid and vitamins B_6_), and vitamin E [22], whereas De Natale et al. showed that this supplemented dairy product failed to change CRP values either in patients with MS [4] or patients with PVD [20]. Additionally, no remarkable alterations in CRP values were observed in patients with mild mixed hyperlipidemia [4], hypercholesterolemic children and adolescents [5], and participants with moderate cardiovascular risk [18], following consumption of the test products containing B vitamins and *omega*-3 PUFAs. The contradictory findings may be due to initial metabolic differences in the study populations, as participants had quite low CRP values in one study at baseline [4] and a more significant beneficial effect may be observed in individuals with higher initial levels, such as patients with diabetes [4]. Moreover, other than different doses and combinations of B vitamins and *omega*-3 PUFAs administered in trials, the addition of other nutrients, such as vitamin C or oleic acid, to the supplementation may have altered the synergistic effect between B vitamins and *omega*-3 PUFAs on CRP [19,22].

TG and LDL-C were positively associated with the incidence of CVD [3,4,5,21,25]. Single supplemental *omega*-3 PUFAs could decrease TG [34], which were commonly observed in adults with hyperlipidemia when it was supplemented at higher doses (ranging from 2 to 4 g/day) [35,36,37]; however, these results were not observed in adults with normal lipid levels, and effects of supplemental *omega*-3 PUFA on LDL-C and HDL-C remain inconsistent. [21,38,39,40,41,42]. More recent studies investigating specific *omega-3* PUFA combinations have shown that reduction in TG levels in adults with high cholesterol are observed with animal-based EPA + DHA supplementation, sole EPA supplementation derived from either fish or microalgae, but not with sole ALA supplementation [43,44]. Additionally, single supplemental B vitamins have also been shown to reduce TG and cholesterol in adults with heart diseases [28].

In the studies included in the present review, the various dosages and combinations of the nutrients may contribute to the heterogeneous effects of the combined supplementation of B vitamins and *omega*-3 PUFAs on TG, LDL-C, and HDL-C. Furthermore, the variations in LDL-C are not concomitant with the changes of TG under co-ingestion of B vitamins and *omega*-3 PUFAs [5,22,24]. The extra ingredients contained in the fortified food may also counteract the effect exerted by the combined supplementation of B vitamins and *omega*-3 PUFAs on LDL-C. In the study by De Natale et al., the enriched functional products decreased plasma TG, but did not change plasma LDL-C [4]. This may be due to *β*-glucan supplemented to the product food, which has the properties of effectively lowering cholesterol and, thus, maybe counterbalancing the untoward impact of *omega*-3 PUFAs on LDL-C [4]. Additionally, the daily intake of the enriched milk fortified with folic acid and *omega*-3 PUFAs resulted in both plasma LDL-C and TG reduction in patients with mild hyperlipidemia [25], participants with moderate cardiovascular risk [18], and patients with MS [21], but only led to LDL-C reduction with no changes in TG in healthy participants [22] and patients suffering from MI [23]. The possible reasons for these disputed results may be attributed to various health statuses in participants and different study designs.

### 4.2. B Vitamins and Omega-3 PUFAs Supplementation and CVD Risk

There was lack of consistent beneficial effects of B vitamins supplementation on CVD risk. This is supported by a meta-analysis including 15 RCTs which showed that null effect was found under B-complex vitamin therapy (vitamin B_6_, folate, or vitamin B_12_ given alone or in combination) on CVD prevention [10]. However, another recent meta-analysis demonstrated that both folic acid (7 RCTs) and B-complex vitamins (12 RCTs) reduced incident stroke [45]. Similarly, the absence of consistent beneficial effects of *omega*-3 PUFAs consumption on the rate of CVD was also observed. A large meta-analysis including 79 RCTs with 12 to 72 months’ duration indicated that increasing intake of EPA and DHA (mainly from supplementation) exerted little or no influence on mortality or cardiovascular health, whereas ALA consumption might slightly decrease CVD risk [34]. Nevertheless, a recent updated meta-analysis including RCTs reported that supplemental marine *omega*-3 FAs decreased coronary risk [46].

Moreover, there was no study included in the present review reporting the effect of the combined supplementation of B vitamins and *omega*-3 PUFAs on primary prevention of CVD. Regarding secondary CVD endpoints, the SU.FOL.OM3 trial involving 2501 patients with a history of CVD (MI, stroke, and unstable angina) treated with B vitamins (5-methyl-tetrahydrofolic acid + vitamin B_6_ + vitamin B_12_) and/or *omega*-3 PUFAs for about 4.7 years, had reported neither B vitamins nor *omega*-3 PUFAs significantly decreased risks of hard coronary events as well as coronary revascularization [6]. This trial also reported that neither B vitamins nor *omega*-3 PUFAs supplementation exerted effects on blood pressure and the rates of other cardiovascular events in patients with a history of established coronary or cerebrovascular diseases [14,17]. Since this RCT with a double-blind placebo-controlled 2 × 2 factorial design, the findings are relatively reliable.

### 4.3. Possible Mechanisms

B vitamins and DHA are interlinked in one-carbon unit metabolism [47,48]. An animal study revealed that the addition of fish oil to a vitamin B_6_ insufficient diet may prevent the reduced synthesis of EPA and DHA [49]. Likewise, insufficient folic acid in diet significantly reduced *omega*-3 PUFAs in plasma and platelets, compared to the rats fed with adequate folate [8,50]. The possible explanation may be due to the enhanced Hcy level induced by folate depletion, which, in turn, contributes to Hcy-associated oxidative stress with more PUFAs oxidized [8,51]. Therefore, alterations in each of the above nutrients can influence Hcy homeostasis, oxidative stress status and methylation reactions [13]. Furthermore, it has been proved that Hcy can induce atherosclerosis through promoting lipoprotein oxidation and endothelial dysfunction, as well as increasing cholesterol synthesis [25]. The increased reactive oxygen species are activators of the transcription factors, such as nuclear factor *kappa* B (NF-*κ*B) [50]. The Hcy-stimulated NF-*κ*B can lead to a pleiotropic response involving up-regulation of endothelial activation factors [47], contributing to early atherosclerosis and lesion initiation [24]. NF-*κ*B activation is also a potent inducer of proinflammatory cytokine [52]. Therefore, B vitamins and *omega*-3 PUFAs may protect against endothelial dysfunction via reduction of proinflammatory cytokines activation through anti-hyperhomocysteinemia and lowering oxidative stress [7,13]. Alternatively, *omega*-3 PUFAs are postulated to inhibit lipogenesis and enhance resolvin and protectin generation, eventually contributing to decreased inflammation [53]. Possible direct adverse effects on atherosclerosis could also appear under high doses of B vitamins, such as folic acid [4]. The possible mechanisms may be because folic acid promotes the remethylation of Hcy to methionine with elevating S-adenosyl methionine, thus resulting in enhancing asymmetrical dimethylarginine concentrations, which may suppress the synthase of endothelial nitric oxide [8,54]. Additionally, these folic acid-induced methylation effect may also influence several proatherogenic genes expression [54].

### 4.4. Implications for Practice

Although the combined supplementation of B vitamins and *omega*-3 PUFAs may have more beneficial impact on CVD prevention than B vitamins or omega-3 PUFAs given alone, dietetic strategies for preventing CVD need to focus more on the importance of considering effects at the whole food and dietary patterns level. Healthy dietary patterns including foods rich in B vitamins and *omega*-3 PUFAs (such as fish, vegetable, fruit, legumes, nuts, and eggs) are likely more useful for preventing or treating CVD, because they provide other beneficial nutrients which are not present in the pure supplements [55,56]. The focus of primary prevention remains rooted in a whole diet approach. For example, the Mediterranean diet, which is rich in *omega*-3 PUFAs and B vitamins, was recently reported to reduce CVD risk by 20–25% in individuals with high adherence to this diet [57]. The synergistically beneficial effects on CVD risk from the possible interactions among multiple nutrients in foods abundant in B vitamins and *omega*-3 PUFAs cannot be ruled out.

### 4.5. Study Limitations and Methodology Challenges

Several limitations need to be considered when interpreting the findings from this review. First, the regimens of nutrient supplementation are different among the human trials, which include capsules, an enriched bake product, an emulsified preparation, and enriched milk with different dosages and combinations of B vitamins and *omega*-3 PUFAs. Additionally, although the capsules allocated to different groups were identical in appearance and smell, as well as taste, the number of total capsules to be taken in Huang et al.’s study was different among groups [3]. Therefore, potential bias may exist, and techniques, such as double dummy, are suggested in future studies to avoid such bias. Second, owing to the absence of randomization and/or a placebo-controlled group in some included trials, the so-called “placebo effect” cannot be excluded. Furthermore, without the placebo-controlled group, interaction effect could not be determined. Therefore, a factorial study design, e.g., 2 × 2 factorial design, is warranted for future studies, which allows the researchers to investigate the interaction between B vitamins and *omega*-3 PUFAs in addition to their respective main effects. Third, participants in most of the included trials maintained the habitual diet. There was no information about the amount of B vitamins and of *omega*-3 PUFAs intake in the usual diet. Fortification status with folic acid in different regions may also influence the effect of the above nutrients’ supplementation on CVD [58]. Thus, the variation in diet could introduce residual confounding that might bias the results of the combined supplementation in the trials especially in the small-scale trials. Fourth, because the participants with CVD received up-to-date pharmacological treatments with nutrient supplementation as an auxiliary means, cardiovascular risk might be further decreased, which possibly masked the potential effects of supplemental B vitamins and *omega*-3 PUFAs [5]. Fifth, there were significant differences in intervention duration, the dosage and formation of the combined supplements across the included studies, which might lead to the inconsistent findings. Moreover, the potential effects of genetic risk factors influencing B vitamins and *omega*-3 PUFAs metabolism were not examined in the included studies. Genetic variations encoding the key enzymes may affect individual’s capacity to utilize/metabolize nutrients [59], thus influencing efficacy of B vitamins and n-3 LC-PUFA supplementation on CVD prevention among diverse population. Additionally, the possible limitations of the present review itself (such as search strategy) cannot be completely ruled out.

## 5. Conclusions

The limited evidence from intervention studies indicates that the combined supplementation with B vitamins and *omega*-3 PUFAs may be promising and more effective at reducing plasma Hcy, TG, and LDL-C than each supplementation alone. However, there is no solid evidence that the joint supplementation of these two can offer a synergistic effect on preventing CVD and decreasing the relevant morbidity and/or mortality in susceptible populations. Due to the methodological challenges and heterogeneity in study design of the existing trials, it is difficult to draw any definitive conclusions based on the current literature. Therefore, well-designed high-quality trials that will use the combined supplementation of B vitamins and *omega*-3 PUFAs or dietary patterns rich in these two types of nutrients are warranted.

## Figures and Tables

**Figure 1 nutrients-14-01608-f001:**
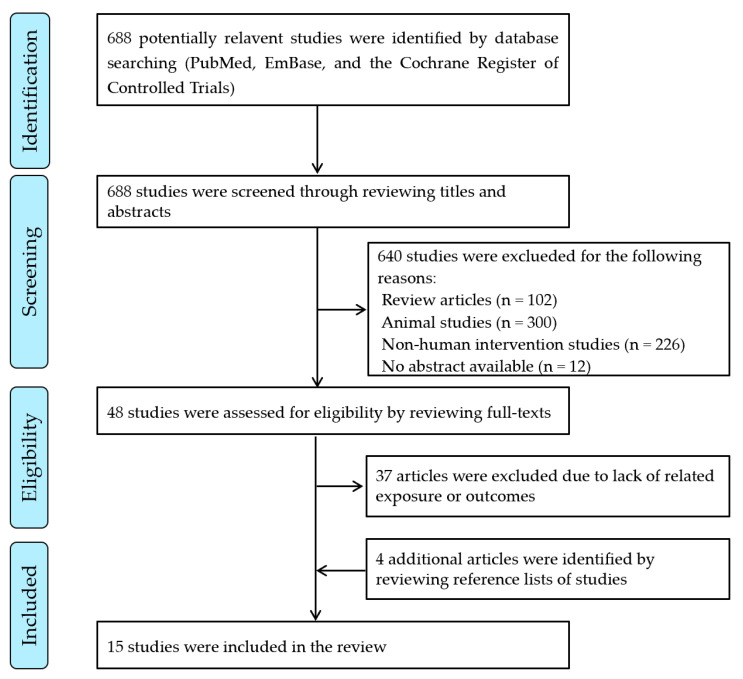
Flow diagram of identification, screening, and selection process for included articles.

**Table 1 nutrients-14-01608-t001:** Characteristics of the studies included in the systematic review.

Reference	Year	Location	Participants	Design	Intervention	Duration	Outcomes
Haglund et al. [12]	1993	Sweden	*n* = 12, male volunteers, healthy or with slightly to moderately increased blood lipids	A double-blind cross-over study	30 mL of fish oil with 30 mL of orange juice/day for 4 weeks followed by a 5-week washout, and then 30 mL of fish oil with 30 mL of orange juice/day supplemented with vitamin B_6_ (80 mg/day) and folic acid (10 mg/day) for 4 weeks. Fish oil contained 13% DHA and 19% EPA.	Fish oil- supplemented orange juice for 4 weeks followed by a 5-week washout period and then B vitamins + fish oil-supplemented orange juice for 4 weeks	↓Atherogenic index (12%) after fish oil alone and (24% ) after B vitamins-supplemented fish oil; ↓Plasma fibrinogen (6%) after fish oil alone and (15%) after B vitamins-supplemented fish oil; ↓TG and↑HDL-C after fish oil with and without B vitamins supplementation; ↓Plasma Hcy (30%) after B vitamins-supplemented fish oil.
Baró et al. [24]	2003	Spain	*n* = 30, healthy volunteers (50% man)	CS without control	500 mL/day of semi-skimmed milk for 4 weeks and then 500 mL/day of the enriched milk (containing *omega*-3 PUFAs, oleic acid, vitamins E, B_6_, and folic acid) for 8 further weeks.	8 weeks	↓Plasma total cholesterol (6%), LDL-C (9%), Hcy level (13%), LDL hydroperoxides (12%), and VCAM-1(16%) after intervention; No change: plasma TG and HDL-C.
Carrero et al. [26]	2004	Spain	*n* = 30, volunteers (50% man) with mild hyperlipidemia	CS without control	500 mL/day of semi-skimmed milk for 4 weeks and then 500 mL/day of the enriched milk (containing *omega*-3 PUFAs, oleic acid, vitamins E, B_6_, and folic acid) for 8 weeks.	8 weeks	↓Plasma TG (24%), total cholesterol (9%), and LDL-C (13%), VCAM-1 (9%) and Hcy (17%) after intervention; No change: Plasma MDA and LDL oxidation.
Carrero et al. [20]	2005	Spain	*n* = 60, patients with PVD and intermittent claudication (100% man)	RCT with parallel- group design	Supplemented group: 500 mL/day of a fortified dairy product containing EPA, DHA, oleic acid, folic acid, and vitamins B_6_, D, A, and E; Control group: 500 mL/day of semi-skimmed milk with added vitamins A and D.	12 months	↓Plasma total cholesterol and ApoB in supplement group; ↓Total Hcy in patients with high initial concentrations; ↑Walking distance before the onset of claudication and ankle-brachial pressure index values in supplement group; No change: plasma MDA, oxidized LDL, TG, CRP, PAI-1, ICAM-1, VCAM-1 and E-selectin.
Benito et al. [21]	2006	Spain	*n* = 72, patients with metabolic syndrome	A randomized, placebo- controlled and open clinical trial of parallel design	Control group: 500 mL semi-skimmed milk/day; Test group: 36 patients consumed 500 mL enriched milk (5.7 g of oleic acid, 0.2 g of *omega*-3 PUFAs, 150 μg of folic acid and 7.5 mg of vitamin E) /day.	3 months	↓Serum total cholesterol (6.2%), LDL-C (7.5%), TG (13.3%), Apo B (5.7%), plasma glucose (5.3%) and Hcy (9.5%); No change: serum insulin levels and plasma CRP.
Carrero et al. [23]	2006	Spain	*n* = 40, male PVD patients	A longitudinal, randomized, controlled study	Group 1: 500 mL/day fortified dairy product containing fish oil, oleic acid, folic acid, vitamins A, D, E, and B_6_; Group 2: 500 mL/day fortified dairy product + simvastatin (20 mg/day); Group 3: 500 mL/day semi-skimmed milk; Group 4: 500 mL/day semi-skimmed milk + simvastatin (20 mg/day).	12 months	Groups 1 and 2 tripled their claudication distance which correlated with ↑ plasma DHA (*r* = 0.40); ↑ ankle-brachial index and ABI in group 2; ↓plasma Hcy in group 1 and 2; ↑total and LDL-C in group 3.
Carrero et al. [22]	2007	Spain	*n* = 40, patients suffered from MI (100% man)	RCT with parallel- group design	Supplemented group: 500 mL/day of a fortified dairy product containing EPA, DHA, oleic acid, folic acid, and vitamins B_6_, D, A, and E; Control group: 500 mL/day of semi-skimmed milk with added vitamins A and D.	12 months	↓Plasma total and LDL-C, apolipoprotein B, and CRP in the supplemented group; ↓plasma total Hcy in both groups, compared to the baseline; No change: plasma HDL-C and TG, heart rate, blood pressure, or cardiac electrocardiographic parameters.
Fonollá et al. [18]	2009	Spain	*n* = 297, participants (84.5% man) with moderate cardiovascular risk	RCT with parallel- group design	Group 1: 500 mL/day enriched milk (containing *omega*-3 PUFAs, oleic acid, vitamins E, B_6_, and folic acid); Group 2: 500 mL/day skimmed milk; Group 3 (control): 500 mL/day semi-skimmed milk.	12 months	↑ Plasma HDL-C (4%) and↓plasma TG (10%), total cholesterol (4%), and LDL-C (6%) under enriched milk consumption. No change: serum glucose, Hcy, and CRP.
Galan et al. [14]	2010	France	*n* = 2501 patients (79.5% man) with CVD (MI, stroke and unstable angina)	RCT with a 2 × 2 factorial design	Group 1: B-vitamins [5-methyl-THF (560 μg), vitamin B_6_ (3 mg) and vitamin B_12_ (20 μg)] and a placebo capsule for *omega*-3 PUFAs; Group 2: *omega*-3 PUFAs (600 mg of EPA and DHA at a ratio of 2:1) and a placebo capsule for B-vitamins; Group 3: both B-vitamins and *omega*-3 PUFAs; Group 4: placebo capsules for both treatments.	4.7 years	↓ plasma Hcy (19%) under B vitamins treatment with no effects on major vascular events [75 vs. 82 patients; HR = 0.90, 95% CI (0.66, 1.23)]; Allocation to *omega*-3 PUFAs had no effect on major vascular events [81 vs. 76 patients; HR = 1.08, 95% CI (0.79, 1.47)].
Szabo et al. [17]	2012	France	*n* = 2501, patients (79.5% man) with a history of CVD (MI, stroke and unstable angina)	RCT with a 2 × 2 factorial design	Group 1: B-vitamins [5-methyl-THF (560 μg), vitamin B_6_ (3 mg) and vitamin B_12_ (20 μg)] and a placebo capsule for *omega*-3 PUFAs; Group 2: *omega*-3 PUFAs (600 mg of EPA and DHA at a ratio of 2:1) and a placebo capsule for B-vitamins; Group 3: both B-vitamins and *omega*-3 PUFAs; Group 4: placebo capsules for both treatments.	4.7 years	↓ plasma Hcy (19%) under B vitamins treatment; No effect of either *omega*-3 PUFAs or B-vitamins supplementation on BP; Change in BP was not associated with change in Hcy.
Earnest et al. [19]	2012	UK	*n* = 100, participants with elevated Hcy (>8.0 umol/L)	RCT with 2 × 2 factorial design	Group 1: placebo; Group 2: MVit (vit C: 200 mg; vitE: 400 IU; vit B_6_: 25 mg; folic acid: 400 ug; vit B_12_: 400 ug) + placebo; Group 3: *omega*-3 PUFAs (2 g *omega*-3 PUFAs, 760 mg EPA, 440 mg DHA) + placebo; Group 4: MVit + *omega*-3 PUFAs.	12 weeks	↓Plasma Hcy under supplementation of MVit, OR = −1.43 [95% CI (−2.39, −0.47)] and MVit + *omega*-3 PUFAs, OR = −1.01, [95% CI (−1.98, −0.04)], compared to placebo and *omega*-3 PUFAs group; ↓Plasma CRP under supplementation of MVit, OR = −6.00 [95% CI (−1.04, −0.15)] and MVit + *omega*-3 PUFAs, OR = −0.98 [95% CI (−1.51, −0.46), but not vs. placebo or *omega*-3 PUFAs group; ↓Plasma TG under supplementation of *omega*-3 PUFAs, OR = −0.41 [95% CI (−0.69, −0.13)] and MVit + *omega*-3 PUFAs, OR = −0.71 [95% CI (−0.93, −0.46)], compared to placebo and *omega*-3 PUFAs group.
De Natale C et al. [4]	2012	Italy	*n* = 16, participants (43.8% man) with mild plasma lipid abnormalities	a randomized crossover design	Group 1: a diet containing baked products enriched with active nutrients [*β*-glucans (3.6 g/day), folic acid (1620 μg/day), long-chain (800 mg/day) and short-chain (400 mg/day) *omega*-3 PUFAs, and tocopherols (120 mg/day); Group 2: a diet containing the same products without active nutrients (control diet).	1 month for each of the control and enriched diets and then cross over to the other diet	↓Fasting plasma TG after the active vs. control diet (1.56 ± 0.18 vs. 1.74 ± 0.16 mmol/L), as was the postprandial level of chylomicron TG and the insulin peak; ↓fasting Hcy (8 ± 0.6 vs. 10 ± 0.8 μmol/L) and hunger feeling at the fifth and sixth hour after active diet; No change: CRP.
Blacher et al. [6]	2013	France	*n* = 2501, patients (79.5% man) with a past history of cardio- or cerebrovascular diseases	RCT with a 2 × 2 factorial design	Group 1: B-vitamins [(5-methyl-THF (560 μg), vitamin B_6_ (3 mg) and vitamin B_12_ (20 μg)] and a placebo capsule for *omega*-3 PUFAs; Group 2: *omega*-3 PUFAs (600 mg of EPA and DHA at a ratio of 2:1) and a placebo capsule for B-vitamins; Group 3: both B-vitamins and *omega*-3 PUFAs; Group 4: placebo capsules for both treatments.	4.2 (± 1.0) years	Neither *omega*-3 PUFAs treatment, nor B vitamins treatment was associated with the occurrence of hard coronary events; Allocation to *omega*-3 PUFAs was not associated with any significant effect on coronary revascularization; B vitamins treatment was associated with 52% increase in the risk of coronary revascularization [HR = 1.52, 95% CI (1.11–2.10)].
Garaiova et al. [5]	2013	Slovakia	*n* = 25, hypercholesterolmic children and adolescents, mean age 16.4 ± 3.8 years	CS without control	A combination of plant sterols esters (1300 mg), fish oil (1000 mg EPA plus DHA) and vitamins B_12_ (50 μg), B_6_ (2.5 mg), folic acid (800 μg) and coenzyme Q_10_ (3 mg).	16 weeks	↓serum total cholesterol, LDL-C, VLDL-C, subfractions LDL-2, IDL-1, IDL-2 and plasma Hcy; ↓α-tocopherol levels after standardisation for LDL-C; ↑*Omega*-3 PUFAs with the dietary supplementation; No change: TG, CRP, HDL-cholesterol and apolipo-protein A1.
Huang et al. [3]	2015	China	*n* = 38, healthy individuals, 57% man, 23 ± 3 years of old	RCT with parallel- group design	Group 1: vitamin B_12_ (1000 μg); Group 2: fish oil (2 g); Group 3: vitamin B_12_ (1000 μg) + fish oil (2 g) (each 1 g capsule provided 490 mg of 22:6 *omega*-3, and 98 mg of 20:5 *omega*-3).	8 weeks	↓Plasma TG, uric acid, CRP, and ferritin after 4 and 8 week supplementation of fish oil, and vitamin B_12_ + fish oil; ↓plasma Hcy by 22%, 19%, and 39% after vitamin B_12_, fish oil, and vitamin B_12_ + fish oil supplementation respectively.

PUFAs, polyunsaturated fatty acids; CVD, cardiovascular diseases; DHA, docosahexaenoic acid; EPA, eicosapentaenoic acid; TG, triglyceride; HDL-C, high-density lipoprotein cholesterol; Hcy, homocysteine; CS, cross-sectional study; LDL-C, low-density lipoprotein cholesterol; VCAM-1, vascular cell adhesion molecule-1; MDA, malondialdehyde; PVD, peripheral vascular disease; RCT, randomized controlled trial; ApoB, apolipoprotein B; CRP, C-reaction protein; PAI-1, plasminogen activator inhibitor-1; ICAM-1, intercellular adhesion molecule-1; ABI, ankle-brachial pressure index; MI, myocardial infarction; THF, tetrahydrofolic acid; HR, hazard ratio; CI, confidence interval; BP, blood pressure; MVit, multivitamins; OR, odds ratio; VLDL-C, very low-density lipoprotein cholesterol; ↓, decrease; ↑, increase.

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
