# Peer review of "Do B Vitamins Enhance the Effect of Omega-3 Polyunsaturated Fatty Acids on Cardiovascular Diseases? A Systematic Review of Clinical Trials"

_nutrients, 2022, doi:10.3390/nu14081608_

Round 1

Reviewer 1 Report

This systematic review aims to examine whether the combined supplementation of both B vitamins and omega-3 PUFAs could provide additional beneficial effects to prevent CVD beyond the effect of each supplement based on clinical trials. Basically, the background was well described and the scientific questions was reasonably raised. The studying methods was good and data correspondingly analyzed correctly. The discussion was appropriately and conclusion carefully drawn.

Minor concerns:

Figure 1 is lack of clarity.  

Author Response

Thank you for your time and comments!

Figure 1 has been revised with higher clarity and updated in the revised manuscript according to the reviewer’s suggestion.

Reviewer 2 Report

This manuscript presents the findings of a study designed to evaluate the impact of combined administrations of B vitamins and omega-3 PUFAs on cardiovascular diseases. The study is interesting, and well written. Below are some general and specific comments that may improve the quality of the manuscript.

  1. Although the authors have shown the diverse effects on the CVD risk factors upon the combined administration of B vitamins and omega-3 PUFAs, it would be appreciated to show the effect of B vitamin or omega-3 PUFAs alone on CVD risks. This is important to let the readers know how B vitamins or omega-3 PUFAs alone affect CVD risks, which help them understand the importance of the effects of these combinations.
  2. It should be born in mind that the interventions with omega-3 PUFAs are not always the pure compounds. As such, the diverse effects from different clinical trials might be partly due to this difference. 
  3. Regarding to the possible mechanisms, the mechanisms of B vitamin or omega-3 PUFAs alone on CVD risks are expected. Moreover, genetic studies converting omega-6 PUFAs to omega-3 PUFAs should be briefly described, in which the role of omega-3 PUFAs (other than fish oils rich in omega-3 PUFAs ) in CVD risks has been elegantly determined. 

Author Response

Thank you for your time and comments! We respectively disagree with the reviewer's comments and the following is our response.

1. The reviewer mentioned “it would be appreciated to show the effect of B vitamin or omega-3 PUFAs alone on CVD risks?" Even though our systematic review focused on the combined effect of B vitamins and omega-3 PUFAs on CVD prevention, we still provided the following statements on B vitamins or omega-3 PUFAs alone on CVD risk in discussion section of our manuscript:

  • Line 164-176 has summarized the effect of either B vitamins or omega-3 PUFAs on blood Hcy, a CVD risk factor.
  • Line 216-226 has summarized the effect of either B vitamins or omega-3 PUFAs on blood TG and LDL-C, another two CVD risk factor.
  • Line 244-255 has summarized the previous findings regarding effects of either B vitamins or omega-3 PUFAs on prevention of CVD.

2. We have specifically listed the amount of omega-3 PUFAs in this manuscript regarding omega-3 PUFAs or fish oil supplementation if this information is available in the included studies. We have also pointed it out in discussion (Line 330-332) that there were significant differences in intervention duration, the dosage and formation of the combined supplements across the included studies, which might lead to the inconsistent findings.

3. The purpose of this systematic review is to fill up the current research gap by providing evidence on the joint effect of B vitamins and omega-3 PUFAs on CVD prevention. Because B vitamins and omega-3 PUFA are interlinked in one-carbon unit metabolism, we conducted in-depth discussion of the possible mechanisms based on this review focus. Extra discussion of B vitamin or omega-3 PUFAs alone may not add additional information on this topic. However, we added some discussion regarding genetic variation affecting the efficacy of these nutrients supplementation on CVD prevention in the limitation section.

Reviewer 3 Report

This review aims to examine whether the combined supplementation of B 65
vitamins and omega-3 PUFAs could provide additional beneficial effects on improving risk 66
factors to prevent CVD beyond the effects of either of them alone and delineate any in- 67
consistencies and limitations of the current intervention studies, through comprehensive 68
analysis of the available literature, and lastly, provide suggestions for future research and 69
dietetic practice or advice.
They concluded:
The limited evidence from intervention studies indicates that the combined supple- 335
mentation with B vitamins and omega-3 PUFAs may be promising and superior at reduc- 336
ing plasma Hcy, TG, LDL-C than each supplementation alone. However, there is no solid 337
evidence that the joint supplementation of these two can offer a synergistic effect on pre- 338
venting CVD and decreasing the relevant morbidity and/or mortality in susceptible pop- 339
ulations. Because of the methodological challenges and heterogeneity in study design of 340
the existing trials, it is difficult to draw any definitive conclusions based on the current 341
literature. Therefore, well-designed high-quality trials that will use the combined supple- 342
mentation of B vitamins and omega-3 PUFAs or dietary patterns rich in these two types of 343
nutrients are warranted.

This manuscript presents important results, is very well presented.

Author Response

We really appreciate your time for reviewing our manuscript!